# Prediction of Parking Space Availability Using Improved MAT-LSTM Network

**Feizhou Zhang** [1]**, Ke Shang** [1] **, Lei Yan** [1,*]**, Haijing Nan** [2] **and Zicong Miao** [2]

[1] Institute of Remote Sensing and Geographic Information System, School of Earth and Space Sciences, Peking University, Beijing 100871, China; zhangfz@pku.edu.cn (F.Z.); shangke@stu.pku.edu.cn (K.S.)

[2] Hybrid and Intelligent Cloud Fusion Innovation Lab, China Telecom Cloud Computing Corporation, Beijing 100083, China; nanhj@chinatelecom.cn (H.N.); miaozc@chinatelecom.cn (Z.M.)

[*] Correspondence: lyan@pku.edu.cn

**Abstract:** The prediction of parking space availability plays a crucial role in information systems providing parking guidance. However, controversy persists regarding the efficiency and accuracy of mainstream time series prediction methods, such as convolutional neural networks (CNNs) and recurrent neural networks (RNNs). In this study, a comparison was made between a temporal convolutional network (TCN) based on CNNs and a long short-term memory (LSTM) network based on RNNs to determine an appropriate baseline for predicting parking space availability. Subsequently, a multi-head attention (MAT) mechanism was incorporated into an LSTM network, attempting to improve its accuracy. Experiments were conducted on three real and two synthetic datasets. The results indicated that the TCN achieved the fastest convergence, whereas the MAT-LSTM method provided the highest average accuracy, namely 0.0330 and $1.102 \times 10^{-6}$, on the real and synthetic datasets, respectively. Furthermore, the improved MAT-LSTM model accomplished an increase of up to 48% in accuracy compared with the classic LSTM model. Consequently, we concluded that RNN-based networks are better suited for predicting long-time series. In particular, the MAT-LSTM method proposed in this study holds higher application value for predicting parking space availability with a higher accuracy.

**Keywords:** parking space availability; intelligent transportation; long short-term memory; temporal convolutional network; deep learning

## 1. Introduction

The prediction of parking space availability plays a crucial role in information systems providing parking guidance. With the advancement and refinement of smart cities and intelligent transportation technology, the demand for more accurate and efficient predictions of traffic flow is growing. In response, these predictions are becoming more refined and specialized. Moreover, the focus of traffic prediction is shifting from flows along main roads to streams within peripheral networks. Dynamic updates and accurate predictions on parking space availability enable users and administrators to make rational decisions regarding traffic flow within parking lots, facilitating efficient and user-friendly parking and promoting the balanced utilization of parking facilities and adjacent roads [1,2].

To address the demand for short-term predictions of parking space availability, this study evaluates the effectiveness and practicality of integrating either a temporal convolutional network (TCN) based on convolutional neural networks (CNNs) or a long short-term memory (LSTM) network based on recurrent neural networks (RNNs) into an intelligent parking information system. The two main contributions of this study are as follows:

1.  The effectiveness and accuracy of two neural network models—TCN and LSTM—were compared to identify the most suitable method for single-input single-output time series prediction problems. For the short-term prediction of available parking spaces,

three real and two synthetic datasets were used to train and test the network models. While the TCN has a simple structure, is rapidly trained, and provides high efficiency, the LSTM network can fully leverage its ability to capture longer historical information features, achieving a greater accuracy in handling long-term series problems.

2.  The LSTM network was optimized by implementing a multi-head attention (MAT) module. The improved MAT-LSTM network effectively models the internal correlations between temporal data and fully explores high-level temporal features. Through ablation experiments, it was proven that, at the expense of permissible time costs, the proposed MAT-LSTM network model successfully captured intrinsic correlations between temporal data and improved the accuracy of short-term parking space availability predictions.

The remainder of this paper is structured as follows. Section 2 discusses the literature related to existing prediction methods. Section 3 describes the different architectures of TCN, LSTM, and the improved MAT-LSTM. Section 4 provides the results and statistics regarding the predicted number of available parking spaces (NAP) obtained with the three networks. Additionally, the results are compared and analyzed, and the advantages and limitations of each approach in this process are described. Section 5 concludes the work, discusses challenges and knowledge gaps, and presents opportunities for future studies.

## 2. Related Work

The essence of parking space availability prediction lies in time series prediction, which forms the basis for constructing predictive models. With the development of machine learning, neural-network-based models for time series prediction have gained increasing attention in recent years. Parking space prediction models using fuzzy neural networks, backpropagation (BP) neural networks [3,4], wavelet neural networks [5], wavelet transforms, and particle swarm wavelet neural networks [6] have outperformed traditional linear time series prediction methods.

Neural network models are more suitable for predicting parking space information, owing to their robustness, fault tolerance, and capability to recognize nonlinear complex systems [7,8]. To date, series modeling against the backdrop of deep learning has involved RNNs. Some scholars argue [9] that, when modeling series data, CNNs can outperform RNNs while circumventing the common drawbacks of recursive models, such as exploding or vanishing gradients and inadequate memory retention.

Within the RNN framework, representative models include LSTM and gated recurrent units (GRUs). An LSTM, equipped with memory cells and gating functions, models long-term dependencies and resolves vanishing gradients, offering significant advantages in time series prediction [10]. It has been widely applied in fields such as stock prediction in financial markets and short-term traffic flow prediction [11–14]. Nevertheless, standard LSTM neural networks encounter problems in time series prediction, such as a high time consumption and complexity [15]. The accuracy of predicting occupancy rates and available parking spaces reaches a mean absolute error (MAE) of 0.067 [16] on standard RNNs and a root mean square error (RMSE) of 5.42 [17] on LSTM models.

CNNs, which were initially developed for image processing, have gained popularity owing to their simple structure and fast computational speed [18]. Recent studies have explored the application of CNNs to solving time series prediction problems. For instance, TCN achieved a mean squared error (MSE) of 0.96 in ultra-short-term single-input and single-output prediction tasks for NAP [19].

After integrating a spatial attention mechanism, the multi-input and single-output A-TCN network achieved an accuracy of 0.0061 MSE in short-term prediction tasks involving congestion indices. Moreover, for small-sample datasets (those with fewer than 400 sets of data), TCN networks can converge very rapidly [19]. Therefore, in tasks requiring network efficiency, CNN-based TCNs are preferred.

To further enhance the performance of time series prediction, researchers have focused on enhancing the capacity of neural networks to capture global information by introducing

attention mechanisms [20]. The attention mechanism, originally proposed for image recognition, enables models to effectively focus on specific local information and extract deeper feature information [21]. Similarly, while processing series data, the attention mechanism allows networks to focus on critical information while ignoring unimportant parts. This technique can enhance model performance, especially when long series of data are used. When provided with the same set of queries, keys, and values, models utilizing attention mechanisms can learn different behaviors based on the same attention mechanism. These different behaviors are then combined as knowledge, capturing the dependencies of various ranges within series (e.g., short-range and long-range dependencies). Therefore, it might be beneficial to allow attention mechanisms to combine different representation subspaces using queries, keys, and values. Bhosale et al. [22] were the first to combine an attention mechanism with an RNN. They calculated alignment–probability matrices for input and output series in an encoder–decoder model, effectively solving machine translation issues. Yin et al. [23] proposed an effective method for using the attention mechanism in CNNs to accomplish machine reading comprehension tasks.

Based on the current research, we identified several questions and explained them in this work:

1. For short-term predictions of NAP, which performs better—RNN with LSTM or the structurally simple CNN? In previous studies, CNNs and RNNs were usually discussed independently. When selecting the network baseline, it is a key prerequisite to discuss the efficiency and accuracy of CNNs and RNNs in comparing these two network architectures on the same dataset, which is missing in previous studies. Moreover, different articles use different evaluation metrics, such as MAE in [16], RMSE in [17], and MSE in [19], which makes it difficult to compare the two frameworks in the same study or through the results of different studies.
2. If an RNN with LSTM outperforms a CNN, what are its advantages over the CNN? Are there further methods available to enhance its advantages?

To answer the above questions, we conducted the following research:

1. We simultaneously trained and tested the TCN and LSTM networks on three real datasets and two synthetic datasets and evaluated their characteristics in terms of training time, accuracy, and convergence rate.
2. We improved the LSTM network by integrating a preceding MAT module to capture the features and relationships in long-time series and compared the predictive performance of the improved MAT-LSTM network with that of the classic LSTM network.

## 3. Proposed Approach

This section discusses three types of network architectures: the classic TCN, the classic LSTM, and the improved MAT-LSTM. The research steps and flow chart are illustrated as Figure 1.

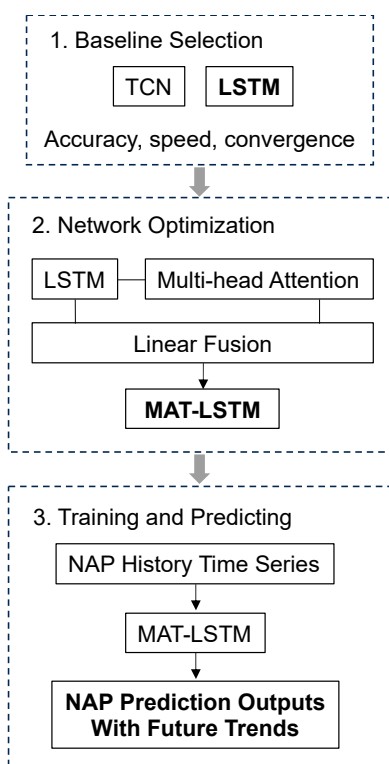

**Figure 1.** The research flowchart and content framework of this paper.

*3.1. Classic TCN Architecture*

As depicted in Figure 2, the TCN model [19] is built upon the CNN architecture and employs causal convolutions to suit sequential modeling. It employs dilated convolutions and residual blocks to capture historical memory. A TCN employs a one-dimensional convolutional network. The TCN architecture employs causal and dilated convolutions, where the value at time t for each layer depends solely on the values of the previous layer at time t, t−1, etc., revealing the characteristics of causal convolution. Moreover, the information extracted from each layer to the preceding layer is skip-connected, and the dilation rate increases exponentially by a factor of two per layer, highlighting the attributes of dilated convolution. Owing to the adoption of dilated convolutions, padding (typically zero-padding) is required for each layer, where the padding size is (kernel size − 1) × dilationrate.

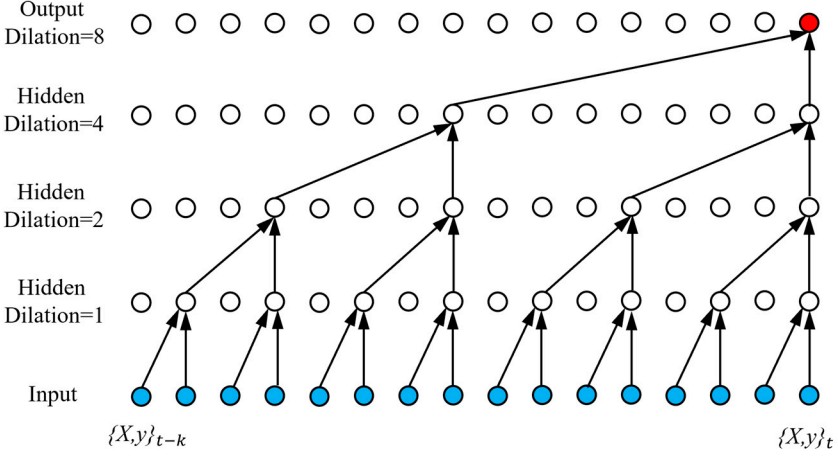

**Figure 2.** TCN architecture.

### 3.2. Classic LSTM Network Architecture

The LSTM [12] network is a variant of an RNN, as shown in Figure 3a. Unlike feed-forward neural networks, RNNs include hidden states that evolve over time. However, traditional RNNs encounter the vanishing gradient problem when trained using backprop-agation through time, rendering them inadequate for handling long-term dependencies within time series data. Thus, an LSTM was introduced to address this issue. As shown in Figure 3b, the additional gated units in the LSTM network enable the retention of long-term patterns from historic time series.

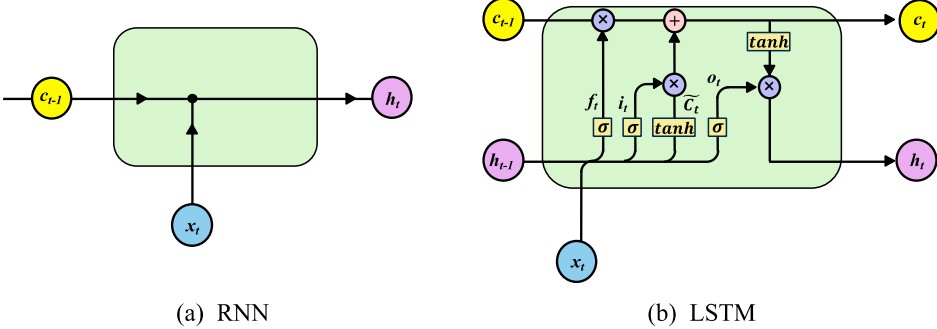

(a) RNN  (b) LSTM

**Figure 3.** Structures of the RNN and the LSTM network.

Formally, the LSTM can be represented as follows: At each time step t, $x_t$ represents an input vector, $c_t$ signifies the memory-state vector, and $h_t$ is the hidden-state vector derived from $c_t$.

The LSTM neural network model implements three gate computations, namely the forget gate, input gate, and output gate, to protect and control the cell state. The specific computational steps and meanings are as follows, where $W_*$ and $U_*$ are weight matrices, $b_*$ is the bias vector, and $\text{sigmoid}(\cdot)$ serves as the activation function for three types of gates. The input modulation $\widetilde{c}_t$ and output $h_t$ often use the hyperbolic tangent $\tanh(\cdot)$ as an activation function. "$\cdot$" denotes the dot-product operation:

1.  Forget Gate Computation: The forget gate decides how much of the cell state from the previous moment is retained in the current cell state, essentially determining what information to discard from the cell state. This gate reads $h_{t-1}$ and $x_t$, then after passing through a sigmoid layer, it outputs a number between 0 and 1, denoted as $f_t$, which is multiplied element wise by each number in the cell state $c_{t-1}$. A value of $f_t$ equal to 0 signifies complete discarding, whereas 1 indicates complete retention. The output of the forget gate is expressed as:

$$f_t = \text{sigmoid}(W_f x_t + U_f h_{t-1} + b_f) \tag{1}$$

2.  Input Gate Computation: The input gate determines how much of the current input is retained in the current cell state and consists of two parts. The first part is a sigmoid layer, which decides which values to update, denoted by $i_t$; the second part is a tanh layer, which creates a new candidate cell state vector, $\widetilde{c}_t$, incorporating the information to be updated into the cell state. The computations for the input gate are expressed as:

$$i_t = \text{sigmoid}(W_i x_t + U_i h_{t-1} + b_i) \tag{2}$$

$$\widetilde{c}_t = \tanh(W_c x_t + U_c h_{t-1} + b_c) \tag{3}$$

3.  Updating the Old Cell State: After processing through the input gate, the old cell state $c_{t-1}$ is updated to $c_t$ by multiplying the old state with $f_t$ to discard the information

that is determined to be discarded and then adding $i_t \cdot \tilde{c}_t$, thus completing the updating of the cell state. It is represented as follows:

$$c_t = f_t \cdot c_{t-1} + i_t \cdot \tilde{c}_t \tag{4}$$

4. Output Gate Computation: The output gate decides how much of the current cell state to output, with a sigmoid layer determining which part of the cell state will be outputted. The cell state is processed through a tanh function to yield a value between $-1$ and 1, which is then multiplied by the output of the sigmoid gate, thus only outputting the determined part of the state. This is represented as:

$$o_t = \text{sigmoid}(W_o x_t + U_o h_{t-1} + V_o c_t + b_o) \tag{5}$$

$$h_t = o_t \cdot \tanh(c_t) \tag{6}$$

When predicting NAP, these three types of gates regulate the information available from the parking spaces entering and leaving the memory cell. The input gate regulates the quantity of new information (e.g., new available parking places) allowed into the memory cell, the forget gate determines how much information to retain in the cell, and the output gate defines the amount of information that can be outputted. The gate architecture of an LSTM enables it to strike a balance between short-term and long-term dependencies in time series data regarding NAP.

### 3.3. Improved MAT-LSTM Network Architecture

The key to successfully completing long-time series prediction tasks lies in thoroughly exploring the deep temporal features and abstracting the inherent relationships within the context. This study proposes an improved MAT-LSTM network that effectively extracts historical temporal features and internal correlations from long-time series and subsequently leverages these properties to enhance its predictive accuracy.

In deep learning models, attention mechanisms are commonly realized by incorporating additional network layers that can learn how to calculate weights and applying these weights to the input signals. Common attention mechanisms include self-attention and MAT, among others. The integration of MAT with neural network models enables the extraction of internal dependencies among elements at different positions within a feature series. This study employs MAT to extract crucial information from long-time series.

As shown in Figure 4, by designing an attention mechanism with h heads, we can independently train h sets of different linear projections to transform the items to be queried (queries) or stored (keys) in the network, as well as specific pieces of information kept in the memory (values). Subsequently, these h sets of transformed queries, keys, and values are concurrently subjected to attention pooling. Finally, the h attention pooling outputs are concatenated and transformed through another learnable linear projection to generate the final output.

In the realm of deep learning, the application of MAT enables the computation of a weighted context vector. This vector is derived by evaluating the similarity between the present and preceding temporal states. Subsequently, this vector is assimilated into the present input, harnessing the principles of self-attention mechanisms. This strategy augments the model's proficiency in discerning complex patterns through the amalgamation of information across various representational subspaces. Significantly, MAT enhances the model's aptitude for focusing on heterogeneous information sources situated at disparate spatial locations, thereby ameliorating its pattern recognition and learning efficacy.

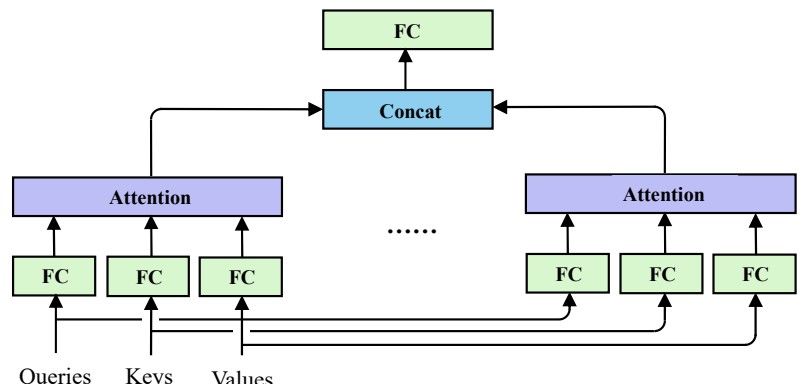

**Figure 4.** Schematic of MAT structure. FC represents a linear transformation, Attention indicates the application of attention -pooling to each head and Concat denotes the concatenation of the *h* attention pooling outputs.

The procedural phases of the MAT consist of the ensuing steps:

1.  The inputs denoted as queries (Q), keys (K), and values (V) undergo linear transformation, reformulating the matrices Q, K, and V, which possess a dimension of $d_{model}$, into correspondingly dimensioned spaces $Q \in R^{m \times d_k}$, $K \in R^{m \times d_k}$, and $V \in R^{m \times d_v}$. This transformation propels the input matrices into more narrowly defined subspaces, thus permitting diverse attention heads to discern distinct facets of the data.

2.  The scaled dot-product attention mechanism is deployed to deduce the outcomes, as depicted in Figure 5. Initially, the dot product of Q and the transpose of K ($K^T$) are computed and then scaled by dividing by the square root of $d_k$ ($\sqrt{d_k}$), a factor that mitigates against the softmax function's propensity for vanishing gradients during training—a common occurrence when dot products are excessively large. Subsequent to the scaling, the softmax function is applied to each row of the scaled scores, engendering a matrix of attention weights that reflect the significance of each corresponding value in relation to each query. Ultimately, the results from the softmax function are multiplied by V, culminating in the output matrix for each attention head, which represents a weighted summation of the values, with the weights mirroring the attention each value receives from the respective queries.

3.  These aforementioned stages are reiterated, culminating in the amalgamation of the respective results. The outputs derived from all attention heads are then concatenated along the dimension of the features to formulate a unified matrix that embodies the information accrued from all heads.

4.  The assembled matrix from step 3 is subjected to an additional linear transformation. The matrix resulting from this process serves as the input for the subsequent layers within the neural network or constitutes the final output in instances where it pertains to the terminal layer in a sequential processing model.

Furthermore, it is noteworthy that the scaled dot-product attention is tantamount to a normalized version of the dot-product attention. In essence, given that the input matrices Q and K are of dimension $d_k$ and V is of dimension $d_v$, the operation executes the matrix multiplication of Q and each K, scales the product by $\sqrt{d_k}$, and thereafter applies the softmax function to ascertain the weights. The output matrix is articulated as follows [22]:

$$\text{Attention}(Q, K, V) = \text{softmax}\left(\frac{Q^T K}{\sqrt{d_k}}\right) V \qquad (7)$$

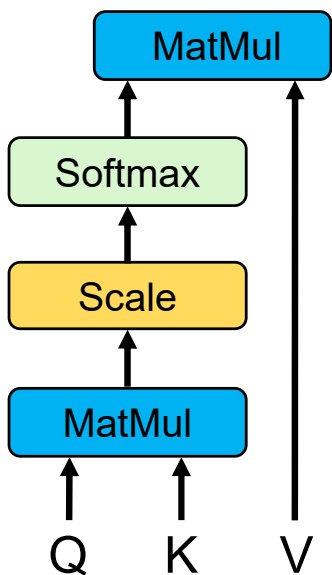

**Figure 5.** Schematic of the scaled dot-product attention structure. Note: MatMul represents the matrix product, Scale denotes normalization through division by $\sqrt{d_k}$, and Softmax calculates the weights.

MAT-LSTM amplifies the model's capability to process sequences in a manner that is both efficient and effective, thereby facilitating concurrent concentration on disparate positions and representational subspaces. This equips the model with an instantaneous and comprehensive grasp of the input data.

## 4. Experiments and Results

This section first introduces the parking lot datasets analyzed in this study, then presents the obtained results, and finally compares the predicted and ground truth values. Additionally, it evaluates and contrasts the performances of the three methods employed in the analysis.

### 4.1. Datasets

In this study, three representative parking lots, labeled D1, D2, and D3, were selected as research subjects. These parking lots are categorized as small- and medium-sized closed off-road facilities, with parking capacities of 157, 132, and 151, respectively. As illustrated in Figure 6, the three parking lots are situated adjacent to each other; however, the data regarding the NAP vary due to differences in geographical locations and surrounding public amenities. Consequently, the three datasets can be cross-referenced.

Among the selected lots, D1 stands out as the busiest, primarily due to its proximity to the subway, resulting in more pronounced fluctuations in NAP, as depicted in Table 1 and Figure A1. D2, located adjacent to D1, shares similar geographical features and transportation accessibility, resulting in minimal variance in the data when D1 diverts the traffic flow. Meanwhile, D3, positioned at a distance from the first two, ranks second in business, owing to its essential function within its locale. Typically, each parking lot operates from 08:00 to 23:59 and closes from 00:00 to 07:59 for maintenance, a period during which no vehicular access is permitted, except on special occasions.

**Table 1.** Data validations and errors.

| Parking Slot | $N_{p,True}$ | $N_{p,Cal}$ | Error |
|:---:|:---:|:---:|:---:|
| D1 | 152 | 152 | 0 |
| D2 | 132 | 132 | 0 |
| D3 | 150 | 150 | 0 |

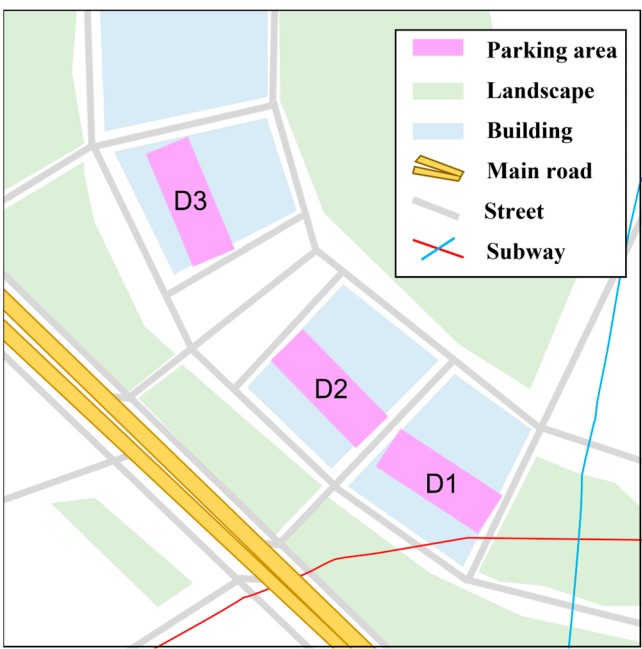

**Figure 6.** Location and surroundings of the analyzed parking lots.

Cameras located at the entrance and exit of each parking lot accurately recorded the arrival and departure times of each vehicle, which were saved as parking records in the parking management system. The original records of the parking lot management system were processed to calculate the NAP in real time by determining the net increase in traffic flow as follows. If the numbers of vehicles passing through the entrance and exit during an interval $\Delta t$ from $t_0$ to $t_1$ are represented by $q_{in,\Delta t}$ and $q_{out,\Delta t}$, respectively, then the net increment in vehicles in the parking lot during such a period is:

$$q_{\Delta t} = q_{in,\Delta t} - q_{out,\Delta t} \tag{8}$$

Given the NAP $N_{t_0}$ in the field at time $t_0$, the NAP $N_{t_1}$ at time $t_1$ can be determined as:

$$N_{t_1} = N_{t_0} - q_{\Delta t} \tag{9}$$

The entrance and exit records analyzed in this study from the three parking lots spanned the entirety of 2021. For each parking lot, indexed as p, the initial NAP at time $t_0$, denoted as $N_{p,t_0}$, corresponds to the total capacity of the parking lot $Q_p$ minus the count of vehicles in the parking lot upon its closure on 1 January 2021, represented as $n_{p,t_0}$, i.e.,

$$N_{P,t_0} = Q_p - n_{p,t_0} \tag{10}$$

To verify the accuracy of the collected data, a verification point was established on 2 October 2021. The actual and the calculated values of NAP, denoted as $N_{p,True}$ and $N_{p,Cal}$, respectively, were compared. $N_{p,True}$ was counted manually in the field during the closure of the parking lot to ensure that the number did not change during the fieldwork. The results of this comparison are presented in Table 1. All NAP data of D1, 2, and 3 in 2021 are shown in Appendix A.

Table 1 indicates that the calculated NAP aligned well with the actual value, demonstrating the accuracy of the preprocessing method. The original data from parking lots D1, D2, and D3 were processed using the proposed algorithm to derive time series data representing the NAP per minute for each parking lot throughout 2021. This process resulted in the formation of datasets D1, D2, and D3, each with a sequence length of 525,600, as depicted in Figure A1. The analysis revealed that, due to the similar geographical locations, attributes, and scales of the three parking lots, the trends, extreme values, and variances

of the datasets were also similar. Therefore, these datasets are representative and able to characterize the overall trend of the NAP in the same types of parking lots in the area. To assess the scalability and applicability of the algorithm, synthetic datasets S5 and S10 were generated as multiples of the summation of the values from the three real datasets plus a set of random numbers, as follows:

$$S5_i = (D1_i + D2_i + D3_i) \times 5 + rand(0, 10) \tag{11}$$

$$S10_i = (D1_i + D2_i + D3_i) \times 10 + rand(0, 50) \tag{12}$$

where $rand(a, b)$ is a random integer between a and b.

The expanded dynamic range of the synthetic datasets allows for the evaluation of the model's performance with larger sample values, considering its potential extension to medium and large parking lots. By incorporating random numbers with varying dynamic ranges into the equation, the model's ability to produce stable outputs across datasets with different degrees of variation is assessed.

The sample standard deviation (SD) is computed as:

$$S = \sqrt{\frac{\sum_{i=1}^{n}(x_i - \bar{x})^2}{n - 1}} \tag{13}$$

where $x_i, i = 1, 2, \ldots, n$ is the sample sequence and $\bar{x}$ is the average of the sequence. SD indicates the degree of dispersion of the data. A larger SD implies a more widely spread distribution of data points and a greater variability within the dataset.

The main characteristics of the datasets are provided in Table 2. An analysis of the table reveals that the dispersion of the real dataset D2 is the lowest, indicating relatively stable data changes. Moreover, the ranges and dynamic ranges of D1, D2, and D3, along with the order of magnitude of the sample values, exhibit similarities, enabling repeated verification to assess the stability of the prediction model. Synthetic dataset S5 exhibits the largest dispersion, whereas synthetic dataset S10 has the largest range. Additionally, the sample values in both synthetic datasets are one order of magnitude higher than those of the real datasets. Consequently, the prediction model can be validated from various perspectives using real and synthetic datasets, facilitating a discussion on its usability in parking prediction tasks.

**Table 2.** Characteristics of the datasets.

| Datasets | Attribute | Min | Max | SD |
|----------|-----------|-----|-----|-----|
| D1 | Real | 0 | 157 | 58.68 |
| D2 | Real | 0 | 132 | 29.00 |
| D3 | Real | 0 | 151 | 42.22 |
| S5 | Synthetic | 1 | 2210 | 1245.83 |
| S10 | Synthetic | 1 | 4450 | 622.89 |

In this study, 4/5 lengths of the total data set were allocated to the training set and the last 1/5 length time series was allocated to the test set, corresponding to days 1–292 and 293–365 of 2021, respectively.

### 4.2. Network Evaluation Method

Employing a unified evaluation metric is essential to ensure the comparability of different methods. In moments of extreme congestion, the NAP may reach zero, rendering proportional functions unsuitable due to the potential for a null denominator. Hence, this study utilizes the mean-squared error (MSE), a widely employed evaluation metric in deep

learning tasks, as the loss function to train and assess the models. The MSE, also referred to as the L2 loss, can be mathematically expressed as:

$$\text{MSE} = \frac{1}{m}\sum_{i=1}^{m}(y_i - \hat{y}_i)^2 \tag{14}$$

where $y_i$ is the input data, i.e., the training set time series; $\hat{y}_i$ is the output data, i.e., the network's predictions; and m is the series length.

### 4.3. Training and Results

The computer utilized for training featured a six-core AMD Ryzen 5 3600 CPU clocked at 3.59 GHz and an RTX 3060 GPU with 12 GB of memory. The software employed included Python 3.8.16, PyTorch 1.8.1, and Cuda 11.1. The model hyperparameters were specified, as shown in Table 3. To mitigate memory consumption during training, particularly due to the large values in the S5 and S10 datasets, the datasets underwent normalization, scaling the time series to values within a range of [0, 1].

**Table 3.** Network hyperparameters.

| Model | Hyperparameter | | | | | |
|---|---|---|---|---|---|---|
| | Batch Size | Epochs | Layers | Hidden Units/Layer | Kernel Size | Learning Rate |
| TCN | 128 | 300/120 | 8 | 30 | 13 | 0.004 |
| LSTM | 128 | 300/120 | 8 | 30 | 13 | 0.0001–0.001 |
| MAT-LSTM | 128 | 300/120 | 8 | 30 | 13 | 0.0001 |

Figure 7 illustrates the NAP predicted by the TCN, LSTM, and MAT-LSTM models, employing the D1 dataset as a case study. All three methods provided accurate NAP predictions, demonstrating similar trends and closely resembling the ground truth. The prediction accuracies and training times across various datasets are listed in Tables 4 and 5, respectively.

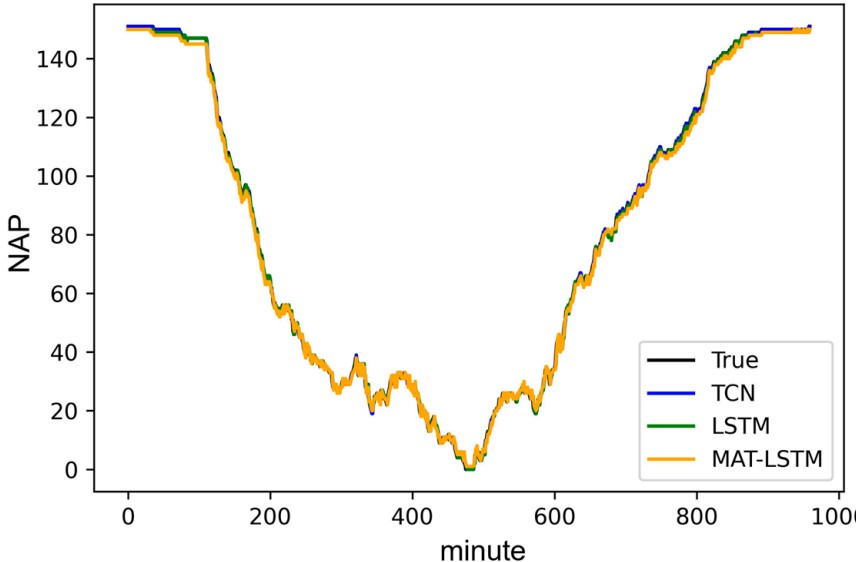

**Figure 7.** True and predicted NAP values for part of the D1 test set.

**Table 4.** Prediction accuracies of TCN, LSTM, and MAT-LSTM on different datasets.

| Model | MSE of the Real Dataset | | | | MSE of Synthetic Dataset | | |
|---|---|---|---|---|---|---|---|
| | D1 | D2 | D3 | Mean of D1&D2 | S5 | S10 | Mean |
| TCN | 0.7903 | 0.3321 | 0.3424 | 0.5612 | 0.0046 | 0.0067 | 0.0057 |
| LSTM | 0.0279 | 0.0592 | **0.0005** | 0.0436 | $2.922 \times 10^{-6}$ | $0.198 \times 10^{-6}$ | $1.560 \times 10^{-6}$ |
| **MAT-LSTM** | **0.0145** | **0.0526** | 0.0321 | **0.0336** | $\mathbf{2.043 \times 10^{-6}}$ | $\mathbf{0.161 \times 10^{-6}}$ | $\mathbf{1.102 \times 10^{-6}}$ |

**Table 5.** Training times of TCN, LSTM, and MAT-LSTM on different datasets.

| Model | Training Time on Real Dataset (300 Epochs) | | | | Training Time on Synthetic Dataset (120 Epochs) | | |
|---|---|---|---|---|---|---|---|
| | D1 | D2 | D3 | Mean | S5 | S10 | Mean |
| TCN | 02:11:49 | 02:12:06 | 02:12:19 | 02:12:05 | 00:53:12 | 00:53:26 | 00:53:19 |
| **LSTM** | **01:56:44** | **01:59:01** | **01:57:10** | **01:57:38** | **00:47:57** | **00:47:59** | **00:47:58** |
| MAT-LSTM | 02:15:14 | 02:18:04 | 02:17:05 | 02:16:48 | 00:55:54 | 00:56:07 | 00:56:01 |

*4.4. Accuracy and Efficiency Comparison between LSTM and TCN*

As indicated in Table 4, both the TCN and LSTM networks attained a high accuracy on both the real and synthetic datasets. The MSE convergence curves for TCN and LSTM are illustrated in Figure 8. Both methods converged rapidly within 50 epochs and stabilized after 100 epochs. This observation suggests the efficacy of both approaches for datasets exhibiting changing trends, similar characteristics, and varying sample sizes.

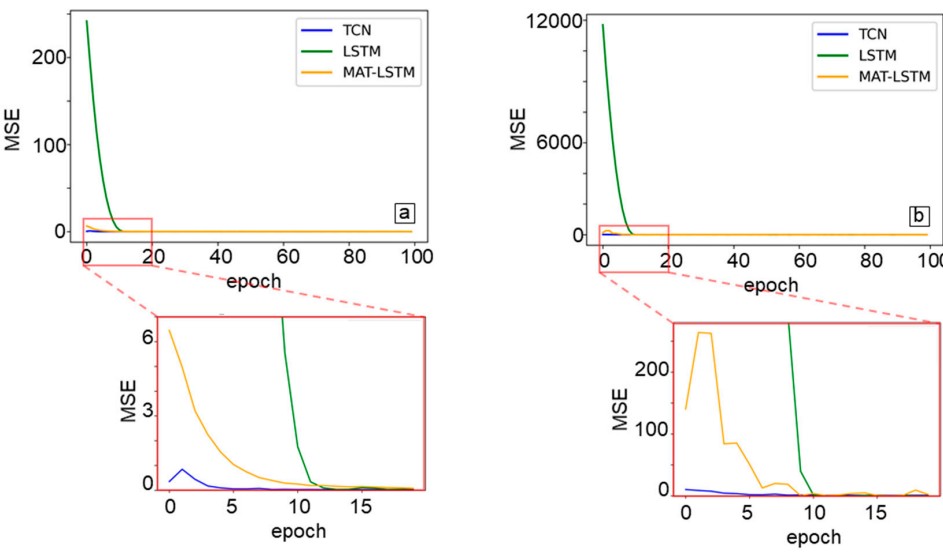

**Figure 8.** Convergence curves for TCN, LSTM, and MAT-LSTM (Dataset D1): (**a**) comparison of validation set MSE curves and (**b**) comparison of test set MSE curves.

Upon a detailed comparison between the accuracy and efficiency of the TCN and LSTM, the following observations were made:

1. LSTM exhibited instability. As previously mentioned, datasets D1, D2, and D3 serve as repeated validations to assess the network stability. If a method performs similarly across these datasets, it has a good stability and repeatability. However, Table 4 reveals a two-order-of-magnitude difference in the MSE of LSTM between the D3 data and D1 and D2 data, suggesting a weak stability. Therefore, the accuracy comparisons below focus solely on the D1 and D2 results, excluding D3.

2. LSTM demonstrated a higher NAP prediction accuracy than the TCN. On the real datasets, LSTM achieved an average MSE of 0.0436, surpassing the TCN's average of

0.5612. On the synthetic datasets, LSTM's average MSE of $1.560 \times 10^{-6}$ exceeded the TCN's 0.0057.

3. Regarding the average training time per epoch, LSTM required less training time than the TCN. Table 5 illustrates that, for datasets comprising 525,600 data points and 300 training epochs, the average training time for LSTM was 01:57:38 compared to that of TCN, which was 02:12:05.

4. LSTM initially exhibited larger errors than the TCN. As depicted in Figure 8, for identical data, the initial MSE of LSTM was two orders of magnitude greater than that of the TCN. Particularly within 10 epochs, the MSE curve of the TCN significantly outperformed that of LSTM. However, after 30 epochs, the slopes of both curves gradually decreased, and the curves converged.

For the data analyzed in this study, it is evident that LSTM achieved a significantly better prediction accuracy than the TCN. However, LSTM needed more training epochs, consistent with the inherent characteristics of RNN and CNN networks. Given that the TCN could develop a model with a certain level of accuracy in a shorter duration, it is suitable for prediction tasks that prioritize time efficiency over stringent accuracy requirements. Examples include rapid training and prediction scenarios involving real-time data analysis.

*4.5. Improvement of MAT-LSTM*

The effect of MAT on the LSTM network was evaluated, and the results of MAT-LSTM were compared with those of LSTM. Based on the preceding analysis, this comparison was not conducted on D3 owing to the unstable LSTM output in this dataset.

1. MAT-LSTM enhanced the prediction accuracy. Table 4 shows that, when comparing the test set MSE of the two networks, MAT-LSTM achieved a significantly higher prediction accuracy on D1 and D2 compared to LSTM. Overall, MAT-LSTM exhibited a 23% reduction in the average MSE compared to the traditional LSTM, indicating a substantial accuracy improvement owing to the network enhancement. In the D1 dataset, MAT-LSTM even achieved a 48% higher accuracy than LSTM. While the accuracy of LSTM was marginally higher than that of MAT-LSTM in the initial stages of training on the S5 and S10 synthetic datasets, after multiple training rounds, MAT-LSTM achieved a final average precision of $1.102 \times 10^{-6}$, surpassing that of LSTM of $1.560 \times 10^{-6}$ by 29%.

2. The convergence speed of MAT-LSTM was accelerated. As depicted in Figure 9, MAT-LSTM attained a lower MSE within the first five epochs and reached a steady state earlier than LSTM on datasets D1, D2, and D3. In contrast, the MSE of the conventional LSTM required 10–40 runs to decrease to a similar level as that of MAT-LSTM. However, on the S5 and S10 synthetic datasets, both networks essentially converged simultaneously, possibly because the normalization process expedited the decline of the loss function. Both networks exhibited a significant downward trend on the S5 and S10 datasets within five epochs and stabilized thereafter.

3. MAT-LSTM required more computational time. The addition of MAT increased the computational workload, increasing the time for each training cycle. As indicated in Table 5, MAT-LSTM consumed 16% more training time per cycle than LSTM for the D1, D2, and D3 datasets, and 20% more for the S5 and S10 datasets.

In conclusion, the enhanced MAT-LSTM model significantly improved the prediction accuracy and compensated for the slow convergence speed of LSTM. However, the introduction of the MAT mechanism entailed increased computational requirements, potentially elevating both time and hardware costs.

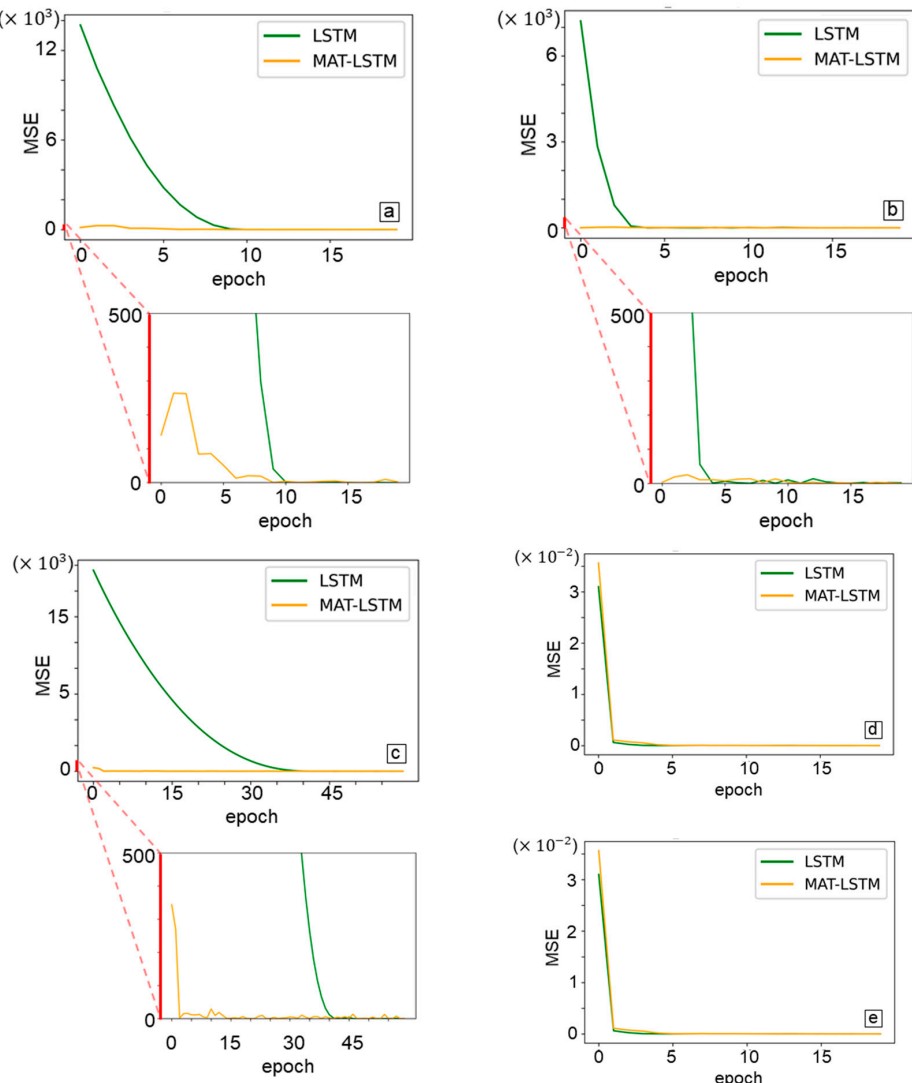

**Figure 9.** Error curves for LSTM and MAT-LSTM: (**a**) Dataset D1; (**b**) Dataset D2; (**c**) Dataset D3; (**d**) Dataset S5; and (**e**) Dataset S10.

## 5. Discussion

In this study, we investigated the performances of two representative neural network architectures, CNN and RNN, in NAP prediction tasks. For long-time series prediction tasks, minimizing the prediction error at each timestep, which may propagate to future timesteps, is crucial.

While RNN-based LSTM demonstrated a higher prediction accuracy, it needed more computational time and resources. Consequently, it is well-suited for periodic system maintenance and error correction. In contrast, the CNN-based TCN model offered high training efficiency and a small memory footprint, making it advantageous for tasks requiring swift qualitative and quantitative predictions, such as dynamic traffic- low prediction, short-term forecasts, and early disaster warnings. The operation of the TCN places less strain on hardware, enabling the completion of training and prediction tasks even on a personal computer with the necessary hardware.

Therefore, considering different application scenarios and target requirements, these two network models, each with their unique strengths, have great potential in the field of traffic flow prediction.

Efforts are ongoing to improve the prediction of parking availability. In future work, the applicability of the algorithm can be explored across various practical aspects, including the impact of time resolution on prediction accuracy. While the dataset used in this

study comprised minute-by-minute time steps, yielding satisfactory prediction results, it is essential to assess the predictive efficacy when applied to datasets with different time step resolutions. For a given time series, a longer step size results in shorter sequence lengths within the same period, capturing more low-frequency information and less high-frequency information. Conversely, shorter time steps lead to denser sampling, capturing more high-frequency information but consuming more storage and computing resources. Thus, in practical applications, time step size is a critical factor influencing system performance, closely tied to the characteristics of the actual data.

Future research can delve into these points:

1.  Within a reasonable range, more universal algorithm models should be explored. The algorithm proposed in this study only relies on historical parking availability time series, which limits the universality of the algorithm model. Specifically, it needs to be trained on the dataset of each parking lot to obtain targeted models. In practical application, our ultimate expectation is a more universal unified model, whose algorithm is applicable to the prediction of parking space availability in different regions and different types of parking lots.

2.  Optimizing the prediction length should be aimed for. Prediction errors tend to accumulate over time, with long-term predictions generally being less accurate than short-term ones. While short-term predictions offer higher confidence in their accuracy, they require more intensive training and computation, potentially leading to redundant computation and resource wastage. By using the same historical data to predict different time lengths, comparing their accuracies and weighing computational requirements, an optimal prediction period can be determined.

3.  Studying a multi-input, single/multi-output network model to explore the impacts of multiple explanatory variables on response variables should be conducted, which can be used to predict the number of highway accidents [24] and incidents of exceeding the bridge design traffic load [25]. In this example, the response variable can be the average availability of a parking lot within a specific time window, and the explanatory variables can include time patterns (hours of the day, days of the week, and holidays [26,27]) and contextual factors (weather conditions [28–30], characteristics of the parking lot in the area, characteristics of the building/enterprise served by the parking lot [31], and so on). This method will simultaneously improve the generalization of predictive models in multiple parking lots, thereby promoting their practical applicability to stakeholders.

4.  Although ANNs are considered "black box" models, the method of using feature importance indicators can effectively rank predictive variables based on their significance, thereby revealing the impact of each predictive variable on the response variable. However, owing to the complexity and variability of transportation, research in this area remains challenging, as the differences in the attributes, user behavior characteristics, and other aspects of each parking lot are significant and may change over time. Therefore, research on this issue should be both targeted and universal, combining the individual attributes of parking lots for analysis and summarizing statistical patterns based on a large number of datasets.

## 6. Conclusions

With the advancement and enhancement of intelligent transportation systems, the demand for NAP prediction is increasing. Effective NAP prediction optimizes the use of urban transportation infrastructure, reducing ineffective and disorderly traffic flow. This study evaluated the accuracy and efficiency of two mainstream neural network models for time-series prediction: the RNN-based TCN and CNN-based LSTM for NAP prediction tasks.

The training and testing results on real and synthetic datasets demonstrated that the RNN-based LSTM network could memorize and use historical time series, achieving a higher accuracy in long-time series prediction tasks. Building upon this foundation, MAT was incorporated into the LSTM network to create MAT-LSTM. This improved network

effectively modeled internal temporal correlations and fully explored high-level temporal features. It achieved an average prediction accuracy of MSE = 0.0336 and $1.102 \times 10^{-6}$ on real and synthetic datasets, respectively. Additionally, it realized average accuracy improvements of 23% and up to 48% (dataset D1) on real datasets and converged faster than LSTM. These results thoroughly illustrate the effectiveness and application potential of the enhanced method.

**Author Contributions:** Conceptualization, Feizhou Zhang and Ke Shang; methodology, Ke Shang and Lei Yan; software, Ke Shang, Haijing Nan and Zicong Miao; validation, Feizhou Zhang, Ke Shang and Haijing Nan; formal analysis, Feizhou Zhang, Ke Shang and Zicong Miao; investigation, Lei Yan and Ke Shang; resources, Feizhou Zhang, Ke Shang and Lei Yan; data curation, Ke Shang, Haijing Nan and Zicong Miao; writing—original draft preparation, Feizhou Zhang and Ke Shang; writing—review and editing, Feizhou Zhang, Ke Shang and Zicong Miao; visualization, Ke Shang and Haijing Nan; supervision, Lei Yan; project administration, Lei Yan. All authors have read and agreed to the published version of the manuscript.

**Funding:** This research received no external funding.

**Data Availability Statement:** MAT-LSTM codes are available at https://github.com/sallyshangke/MAT-LSTM.git (accessed on 24 April 2024). The authors are not authorized to disclose and share the data used in this project. Those interested in using partially erased data for academic research may contact Ke Shang at shangke@stu.pku.edu.cn or sallysk@163.com.

**Acknowledgments:** This study was supported by Peking University and the China Telecom Cloud Computing Corporation. The authors appreciate the helpful discussions with Chenchen Jiang from Peking University and Shupeng Wang from the China Telecom Cloud Computing Corporation during the writing phase of this work.

**Conflicts of Interest:** Author Haijing Nan and Zicong Miao were employed by China Telecom Cloud Computing Corporation. The remaining authors declare that the research was conducted in the absence of any commercial or financial relationships that could be construed as a potential conflict of interest.

**Appendix A**

The NAP curves of D1, D2, and D3 are depicted in Figure A1. Remarkably, the curves from July 21 to 23 exhibit differences compared to those of other periods due to a severe meteorological disaster. Its adverse impact persisted, affecting travel volume into August, thereby resulting in a notable discrepancy between the curves during this period and those of other times.

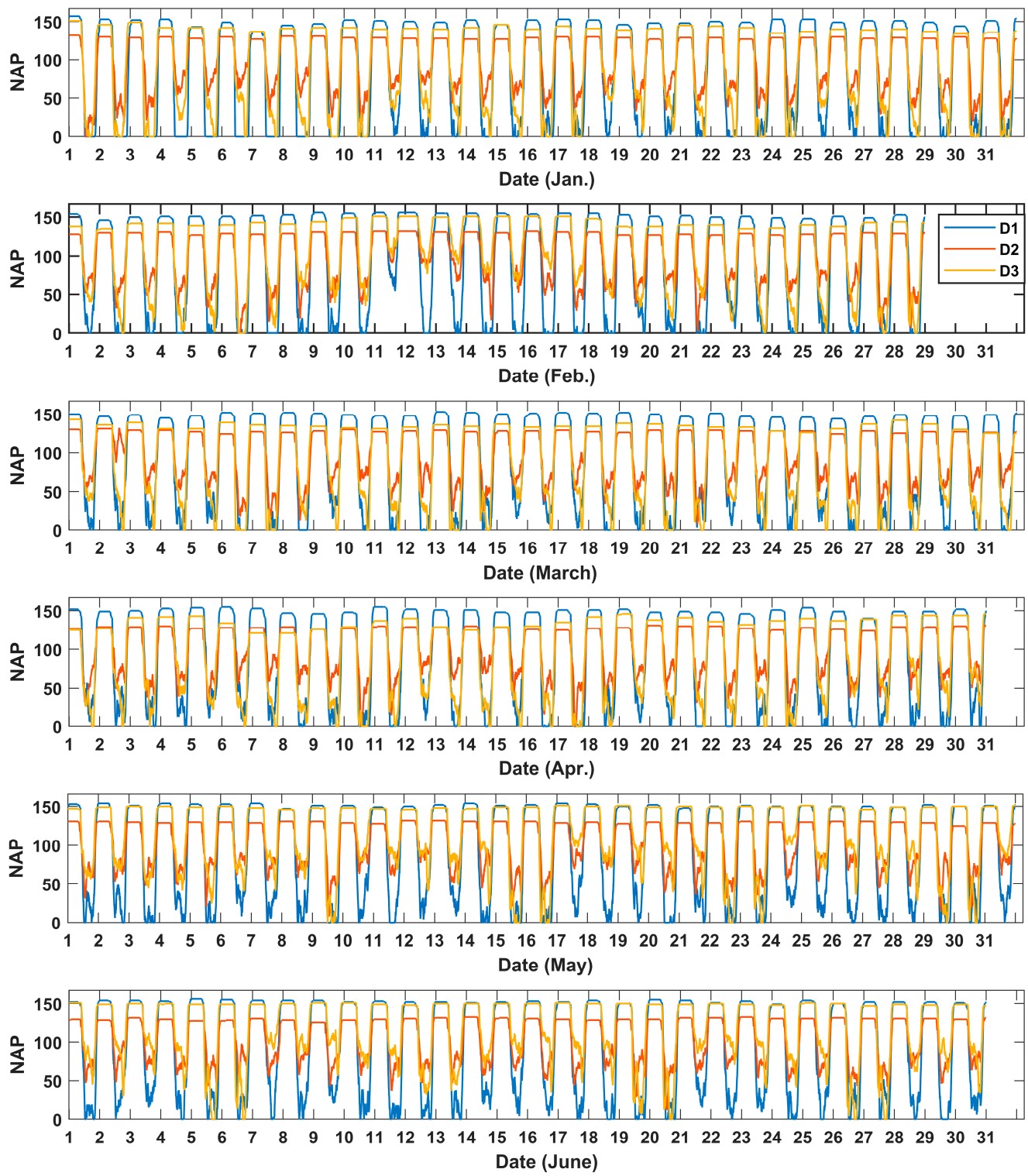

**Figure A1.** *Cont.*

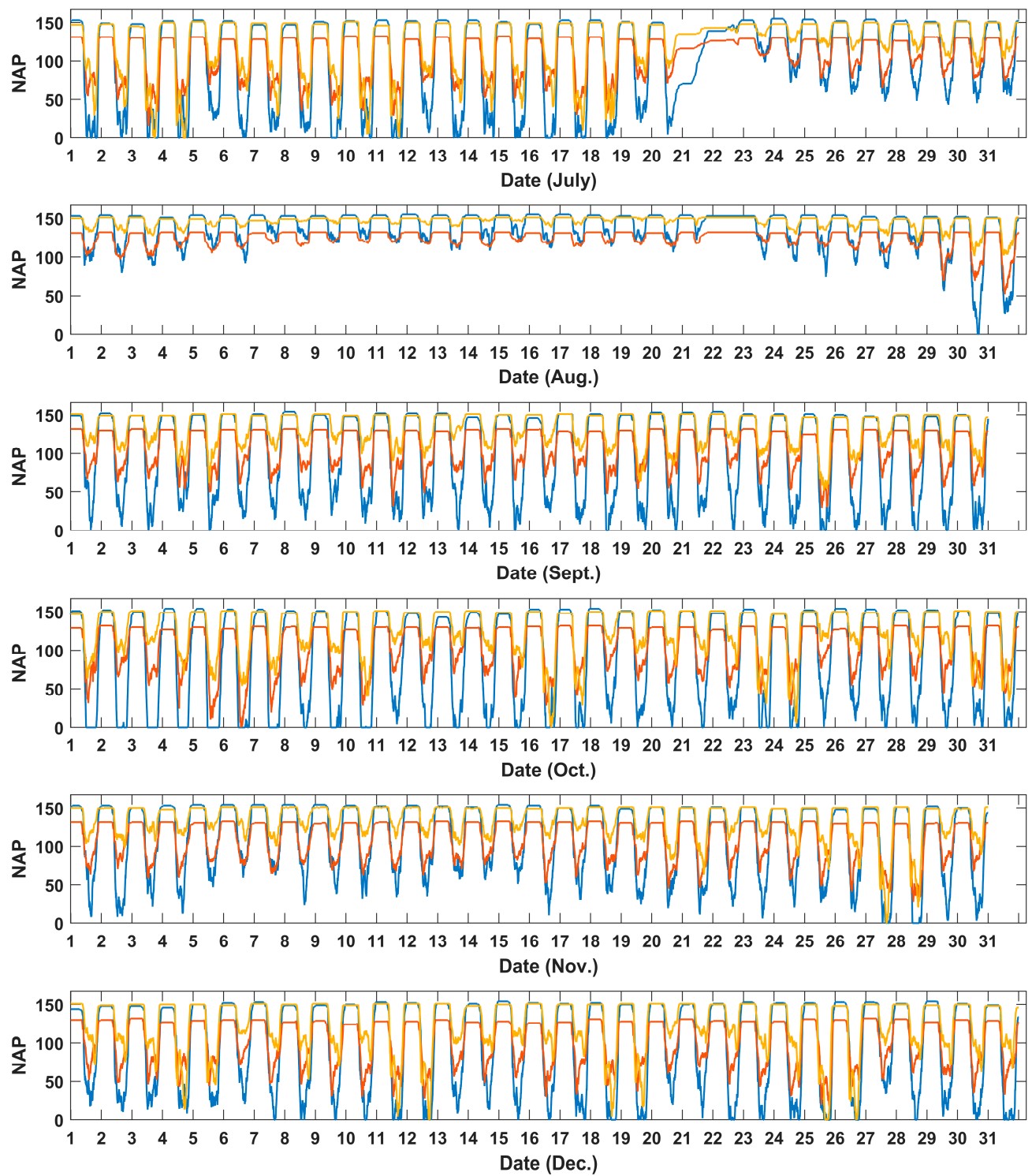

**Figure A1.** NAP of D1, D2, and D3 in the year 2021.

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
