# Peer review of "Prediction of Parking Space Availability Using Improved MAT-LSTM Network"

_ijgi, doi:10.3390/ijgi13050151_

Round 1
Reviewer 1 Report (Previous Reviewer 3)
Comments and Suggestions for Authors
The authors have made a significant improvement in their resubmitted manuscript and have adequately addressed the concerns and suggestions. I have no further comments, hence recommend the manuscript for publication.
Author Response
We appreciate the reviewer’s positive and encouraging comments on the resubmitted manuscript. We are glad to know that the improvements we made have addressed the concerns and suggestions effectively. We appreciate your recommendation for publication and value your confidence in our work. Thank you once again for your rigorous review.
Reviewer 2 Report (New Reviewer)
Comments and Suggestions for Authors
Thank you for the opportunity to review this paper. The research is very interesting and deserves appropriate visibility in this journal. However, I have some comments regarding certain issues thath should be addressed and some suggestions that I hope can further improve this work.
Comment 1
At the beginning of the “3. Proposed Approach” section, I suggest providing a brief flowchart summarising the methodology followed in this research. This could significantly enhance the clarity and readability of your presentation, thereby increasing the impact of your work.
Comment 2
In the "4. Experiments and Results" section, the authors should clarify which tools were used to record the incoming and outgoing flows from the parking lots. Were magnetic loops used? Traffic cameras? Or something else? Additionally, the authors validated the data during a single day (October 2, 2021). What were the weather conditions like on that day? Since the processed data covers an entire year (2021), is it reasonable to assume that only a single day is representative of the entire period? For instance, during adverse weather conditions, I imagine that the accuracy of the measurement of incoming and outgoing flows could decrease. Please discuss this critical point.
Comment 3
In this study, the authors addressed the issue of parking space availability prediction using Convolutional Neural Networks (CNNs) and Long-Short Term Memory (LSTM) neural networks. On the one hand, this is a reasonable and natural approach, as it allows for forecasting future parking space availability based solely on current availability data. On the other hand, one of the main drawbacks of this approach is its lack of generality. Specifically, it requires calibrating a specific model for each individual parking lot. In contrast, other types of neural networks, such as Multi Layers Feedforward Artificial Neural Networks, can predict a response variable based on multiple explanatory variables. In this case, the response variable could be the average availability in a certain parking lot during a specific temporal window, while the explanatory variables could include both temporal factors (hour of the day, day of the week, month, etc.) and contextual factors (weather conditions, characteristics of the area where the parking lot is located, characteristics of the buildings/businesses served by the parking lot, etc.). This approach would allow for a strong generalisation of the predictive model across multiple parking lots simultaneously and thus facilitate its practical applicability for stakeholders. Moreover, although ANNs are considered “black box” models, methods employing feature importance indicators can efficiently rank predictors based on their significance, thus shedding light on the influence of each predictor on the response variable. I believe the authors should discuss this topic as a future development, also including some recent literature works that have utilised Multi Layers Feedforward Artificial Neural Networks to forecast temporally dependent events (e.g., bridge overloads, road accidents) in several fields of transportation engineering (e.g., https://doi.org/10.1016/j.heliyon.2023.e23374; https://doi.org/10.1016/j.trpro.2021.11.038).
Author Response
Dear reviewer,
We appreciate your helpful suggestions, and please refer to the attachment for our responses.

Reviewer 3 Report (New Reviewer)
Comments and Suggestions for Authors
The provided manuscript analyzes the problem of parking space availability forecast and uses real data to compare and test extensively two possible solution methods. The problem is clearly described, and the choice of the tested solution methodologies is motivated correctly. The related literature is quite well analyzed and presented. The paper is very well written and reads smoothly, up to Section 4. The rest of the text should be revised by a native speaker.
However, I have concerns about the scientific contribution of the manuscript. Both analyzed methodologies have been used for time series predictions in various domains. Also, the integration of attention mechanism in LSTM has already been done. Furthermore, the text does not show a valid approach for validating models. Hence, for the paper acceptance, it is essential that the authors address the issues listed below.
Major issues, which must be addressed before the paper acceptance:
1) P2, L90-92: “Recent studies have explored the application of CNNs to solving time series prediction problems. For instance, the TCN achieved a mean squared error (MSE) of 0.96 in ultra-short-term single-input and single-output prediction tasks for NAP.” – Please provide citation as this is an important fact for justifying the usage of TCN in your work.
2) P3, L119-121: “While few previous studies have compared the efficiency and accuracy of these two network architectures on the same dataset, some studies have used partially dissimilar evaluation metrics.” – Kindly explain how your work provides an additional contribution compared to the works described in this sentence. Also, please cite the works you are referring to.
3) In Section 3.3, the authors try to give a brief explanation of multiheaded attention mechanism, but it is not clear enough in my opinion. Firstly, all boxes of Figure 3 should be explained. Then, the authors mention queries, keys, and values, but they do not explain these concepts. Finally, some variables are not defined, like, in particular, Q, K, and V. Since the authors propose MAT-LSTM as their architecture of choice, it should be described in detail, without skipping any notion, and related to the solved problem, in terms which network elements solve which part of the parking availability forecasting problem.
4) P7, L251: Is it not clear what N_(p, Calc) represents and how it is calculated? If this is just counting of the vehicle entries and exits, this validation is too obvious and does not contribute to the paper quality.
5) P11, L342-353: The following two excerpts are contradictory: 1) “LSTM requires less training time than TCN.” and 2) “For the data analyzed in this study, it is evident that LSTM achieves significantly better prediction accuracy compared to the TCN network. However, LSTM demands more training time…” Kindly revise this part of the text to make clear and consistent statements.
6) It is not clear which datasets are used for validation and testing of the models. Is it the same dataset as the one used for training? If so, the calculated errors do not represent the actual model efficiency.
Minor shortcomings:
1) In Figure 3, should it not be “Concat” instead of “Contact”?
2) P8, L263: “Consequently, these datasets serve as a means to assess…” – Please remove the indefinite article.
3) There are misalignments in the used terminology. Up to Section 4, the term “parking” is used, while in Section 4, the term “berth” is used instead. I suggest using the same terminology throughout the text and having the text revised by a native speaker.
Comments on the Quality of English Language
The paper is very well written and reads smoothly, up to Section 4. The rest of the text should be revised by a native speaker.
Author Response
Dear reviewer,
We appreciate your helpful suggestions, and please refer to the attachment for our responses.

Round 2
Reviewer 2 Report (New Reviewer)
Comments and Suggestions for Authors
I sincerely thank the authors for adequately responding to all my previous comments.
Author Response
We appreciate the reviewer for the helpful suggestions for this work.
Reviewer 3 Report (New Reviewer)
Comments and Suggestions for Authors
I thank the authors for addressing the manuscript according to my comments. At this point, I can recommend the manuscript acceptance. I still have two minor suggestions, given below, but I leave it to the authors to decide whether to address them.
Minor shortcomings:
1) The LSTM and MAT-LSTM architectures (Sections 3.2 and 3.3) should rather be explained through ideas, motivations and key concepts, in a descriptive manner, similar to how the classic TCN architecture is presented in Section 3.1. Currently, LSTM and MAT-LSTM architectures are presented in the math notation, while the accompanying description is too brief, so it is not easy to grasp it unless the reader is already familiar with the topics.
2) In Figure 4, the box is still named “Contact” instead of “Concat”.
Author Response
We appreciate the reviewer for the suggestions for this work.
And pleases find our revision for 3.2 and 3.3 according to your advise in the main text, which is marked in blue.
We apologize for incaution of Figure 4. And it was modified in the main text.
This manuscript is a resubmission of an earlier submission. The following is a list of the peer review reports and author responses from that submission.
Round 1
Reviewer 1 Report
Comments and Suggestions for Authors
The comments and review of the paper can be found in the attached file.

Reviewer 2 Report
Comments and Suggestions for Authors
It is not clear what the temporal resolution of NAP datasets (3.1) is used per day. Is it hourly, bi-hourly, or at another interval? Moreover, how does this temporal resolution impact the performance of the proposed model? How are the datasets acquired? Are they accurate enough since part of them are used for accuracy assessment?
In section 3.1, the authors introduced Eq. 8 and 9 to generate synthetic datasets S5 and S10, respectively. While the equations are presented, the rationale behind their formulation and the choice of parameters need to be elaborated and justified. Additionally, the authors opted for (0, 5) to introduce noise in Eq. 8, whereas (0, 100) was utilized for Eq. 9. Suggest clarifying the rationale behind this decision. This would help in understanding the significance of the chosen noise levels and how they impact the synthetic dataset generation differently in each case.
Figures are challenging to decipher. Suggest using alternative symbols or patterns in addition to colors to improve their legibility. Remove the discrepancy in the ordering of figures. For example, two instances are labeled as Figure 5.
Reviewer 3 Report
Comments and Suggestions for Authors
The manuscript, titled ‘A Prediction Method for Available Parking Spaces Based on Improved MAT-LSTM Network’ compares the efficiency and accuracy of parking space prediction methods including CNN-based TCN method and RNN-based LSTM method. Moreover, the study also proposes a MAT-LSTM approach to improve the prediction accuracy. The study has been conducted using 3 real datasets and 2 synthetic datasets. Overall, the research is interesting. However, some issues in the study compel me to not accept the current state of the work. Authors are advised to improve the study by addressing the following points:
1. The structure of the introduction section must be improved. The current structure gives the perception of a technical report or thesis. Sub-headings should be removed. Authors are advised to present the content appropriately for a journal article. Moreover, a separate section for ‘Related work’ may be added after ‘Introduction’ section.
2. The title of the manuscript should not contain any abbreviation. Moreover, abbreviations must be defined at the first instance in the manuscript e.g. abbreviation ‘MSE’ is not defined.
3. Page 2 Line 74: citation is incorrect
4. Figure numbering should be corrected. Figure 5 occurs twice in the manuscript.
5. Background or justification with citations may be added to validate the formation of Synthetic datasets. Why the specific factors of 5 and 10 for S5 and S10, respectively, are used in equations 8 and 9? Please justify.
6. There are a lot of contradictions in results presented in Table 3 & 4 with respect to the detailed comparison presented in the following paragraphs. For example, the paragraph starting at line 302 is not in line with Table 3 and 4. This requires detailed clarification.
7. Similarly, section 3.5 is also quite confusing with respect to the results presented in table 3 and 4.
8. Why only D1 and D2 datasets are being focused; especially while comparing MAT-LSTM with traditional LSTM? Moreover, as per Table 3, average MSE of LSTM is less than that of MAT-LSTM, however the statement at line 318-319 is opposite. There is a lot of ambiguity in the results presented in the manuscripts. Detailed clarification is required to be added.
9. Discussion section should be added. In the current manuscript, some parts included in the results section shall be more relevant in the Discussion section. Also include some limitations of the current approach, methods and the available data.
10. Conclusion should include some prominent quantitative results/outputs.
11. There is no mention of the future directions or recommendations of the research.
12. Lastly, moderate editing of English language required. Numerous typos also need to be corrected.
Comments on the Quality of English LanguagePlease see detailed comments above.